# Promoting Effect of Basic Fibroblast Growth Factor in Synovial Mesenchymal Stem Cell-Based Cartilage Regeneration

**DOI:** 10.3390/ijms22010300

**Published:** 2020-12-30

**Authors:** Gensuke Okamura, Kosuke Ebina, Makoto Hirao, Ryota Chijimatsu, Yasukazu Yonetani, Yuki Etani, Akira Miyama, Kenji Takami, Atsushi Goshima, Hideki Yoshikawa, Takuya Ishimoto, Takayoshi Nakano, Masayuki Hamada, Takashi Kanamoto, Ken Nakata

**Affiliations:** 1Department of Orthopaedic Surgery, Osaka Rosai Hospital, 1179-3, Nagasonecho, Kita-ku, Sakai, Osaka 591-8025, Japan; gensuke-okamura@umin.ac.jp; 2Department of Musculoskeletal Regenerative Medicine, Osaka University Graduate School of Medicine, 2-2 Yamadaoka, Suita, Osaka 565-0871, Japan; 3Department of Orthopaedic Surgery, Osaka University Graduate School of Medicine, 2-2, Yamadaoka, Suita, Osaka 565-0871, Japan; makohira777@gmail.com (M.H.); y_etani@hotmail.co.jp (Y.E.); a-miyama@umin.ac.jp (A.M.); takami-minoh@umin.ac.jp (K.T.); smbc1416@yahoo.co.jp (A.G.); 4Sensory and Motor System Medicine, Graduate School of Medicine, the University of Tokyo, 7-3-1, Hongo, Bunkyo-ku, Tokyo 113-0033, Japan; r-chiji@umin.ac.jp; 5Department of Orthopaedic Surgery, Hoshigaoka Medical Center, Japan Community Health Care Organization, 4-8-1 Hoshigaoka, Hirakata, Osaka 573-8511, Japan; yonechanosk@gmail.com (Y.Y.); hamada-m@umin.net (M.H.); 6Department of Orthopaedic Surgery, Toyonaka Municipal Hospital, 4-14-1 Shibaharacho, Toyonaka, Osaka 560-8565, Japan; yhideki@chp.toyonaka.osaka.jp; 7Division of Materials and Manufacturing Science, Osaka University Graduate School of Engineering, 2-1 Yamada-oka, Suita, Osaka 565-0871, Japan; ishimoto@mat.eng.osaka-u.ac.jp (T.I.); nakano@mat.eng.osaka-u.ac.jp (T.N.); 8Department of Health and Sport Sciences, Osaka University Graduate School of Medicine, 2-2 Yamadaoka, Suita, Osaka 565-0871, Japan; takanamoto@gmail.com (T.K.); ken-nakata@umin.ac.jp (K.N.)

**Keywords:** synovial mesenchymal stem cells, basic fibroblast growth factor, cartilage regeneration, xenograft

## Abstract

Synovial mesenchymal stem cell (SMSC) is the promising cell source of cartilage regeneration but has several issues to overcome such as limited cell proliferation and heterogeneity of cartilage regeneration ability. Previous reports demonstrated that basic fibroblast growth factor (bFGF) can promote proliferation and cartilage differentiation potential of MSCs in vitro, although no reports show its beneficial effect in vivo. The purpose of this study is to investigate the promoting effect of bFGF on cartilage regeneration using human SMSC in vivo. SMSCs were cultured with or without bFGF in a growth medium, and 2 × 10^5^ cells were aggregated to form a synovial pellet. Synovial pellets were implanted into osteochondral defects induced in the femoral trochlea of severe combined immunodeficient mice, and histological evaluation was performed after eight weeks. The presence of implanted SMSCs was confirmed by the observation of human vimentin immunostaining-positive cells. Interestingly, broad lacunae structures and cartilage substrate stained by Safranin-O were observed only in the bFGF (+) group. The bFGF (+) group had significantly higher O’Driscoll scores in the cartilage repair than the bFGF (−) group. The addition of bFGF to SMSC growth culture may be a useful treatment option to promote cartilage regeneration in vivo.

## 1. Introduction

Articular cartilage has a poor self-healing ability and the conventional surgical treatments and cell therapies for cartilage injuries, such as bone marrow stimulation, autologous osteochondral transplantation, and autologous chondrocyte implantation (ACI), have not yielded full recovery of normal cartilage [1]. Furthermore, ACI requires two-step surgery and the sacrifice of normal cartilage tissue to supply new cartilaginous tissue to the defect. Therefore, cartilage regeneration using mesenchymal stem cells (MSCs) has been widely researched [2,3]. Synovial MSCs (SMSCs) are an attractive cell source for cartilage regeneration because they are easily collected as surplus tissue during joint surgery and show favorable chondrogenesis and proliferation potential both in vitro [4,5] and in vivo [6,7]. During the development of commercialization of an allogenic SMSC bank for cartilage repair, we have reported that SMSCs obtained even from young patients showed a large variation in their proliferation and cartilage differentiation potentials [8]. In fact, our human clinical trial revealed that autologous SMSC implantation for the repair of a cartilage defect failed to recover full articular cartilage [9]. Therefore, various pretreatment methods to enhance the cartilage differentiation potential of MSCs have been reported [10,11,12]; we showed that a low oxygen tension prevented cellular senescence and promoted the proliferative and chondrogenic differentiation capacity of human SMSCs in vitro [13]. However, none of these reports succeed to demonstrate the positive effects in vivo. The basic fibroblast growth factor (bFGF) has been widely reported to enhance the potential of proliferation and cartilaginous differentiation of MSCs in vitro [14,15,16]. Studies of embryonic, neural, and bone marrow stem cells reported that bFGF regulates undifferentiated state and multipotent differentiation of stem cells through the fibroblast growth factor receptor 3 (FGFR3)-mediated signaling [14,17,18]. FGFR3 is known as one of the markers of mesenchymal pre-cartilaginous stem cells and was detected at the margins of the cartilage nodules of synovial chondromatosis and SMSCs [19,20,21]. However, no positive effects have been documented in vivo. We hypothesized that SMSCs pretreated with bFGF may demonstrate enhanced cartilaginous differentiation in vivo. The purpose of this study was to investigate whether bFGF can promote cartilage repair using human SMSCs in vivo.

## 2. Results

### 2.1. Proliferative Capacity and Cellular Morphology of SMSCs Cultured in the Presence or Absence of bFGF

SMSCs of the bFGF (+) group showed a shrunken cytoplasm and enhanced proliferation compared with those of the bFGF (−) group (Figure 1a). In addition, the bFGF (+) group maintained higher growth rates than the bFGF (−) group (Figure 1b). The relative proliferation rate of SMSCs from the bFGF (+) group compared with that from the bFGF (−) group was significantly higher from passages 1 to 3 [P1: bFGF (−) vs. bFGF (+), 1.0 ± 0.18 vs. 2.3 ± 0.24-fold; *p* = 0.0075; P2: bFGF (−) vs. bFGF (+), 1.0 ± 0.23 vs. 2.8 ± 0.62-fold; *p* = 0.0075; P3: bFGF (−) vs. bFGF (+), 1.0 ± 0.19 vs. 2.6 ± 1.0-fold; *p* = 0.0075]. However, the difference between the two groups became non-significant after passage 4 (Figure 1c).

### 2.2. Expression of FGFR3 Protein

To analyze the expression of FGFR3 in the SMSCs from different patients, Western blotting was conducted using anti-FGFR3 antibody. The FGFR3 protein expression was confirmed in all of the patients’ SMSCs by Western blotting (Figure 2).

### 2.3. Expression of Cell Surface Markers

To evaluate the effects of bFGF on the expression of cell surface markers in SMSCs, fluorescence-activated cell sorting (FACS) was conducted for four mesenchymal surface markers (CD44, CD73, CD90, and CD105), two mesenchymal negative markers (CD11b, CD271), and one chondrocyte surface marker (CD151) (Figure 3). Almost 100% of the SMSCs expressed all four mesenchymal surface markers and one chondrocyte marker, and almost none of SMSCs expressed mesenchymal negative markers. Moreover, their expression levels were not affected by bFGF.

### 2.4. The Effect of bFGF on the Chondrogenesis of SMSCs in a Three-Dimensional (3D) Chondrogenic Culture

To assess the effect of the addition of bFGF in the cell-proliferation phase on cartilaginous tissue formation, a 3D pellet culture was established. After four weeks of chondrogenic induction, histology, the expression of cartilage differentiation-related genes, and extracellular matrix (ECM) production, were evaluated. Synovial pellets of the bFGF (+) group grew larger and were more strongly stained with Safranin-O (Saf-O) and immunostained for collagen type II (COL2) than those of the bFGF (−) group (*n* = 5 per group) (Figure 4a). A similar tendency was observed in all patients. Moreover, pellets of the bFGF (+) group showed hyaline cartilage-like differentiation with chondrocyte-like round-shaped cells in the lacuna, abundant collagen stained with Sirius Red and abundant ECM stained with Saf-O and Alcian Blue (Figure 4b).

To assess the effect of pretreatment with bFGF on cartilaginous tissue formation of SMSC in vitro, a quantitative evaluation of the cross-section area (CSA) of pellets demonstrated that pellets of the bFGF (+) group grew significantly larger than those of the bFGF (−) group (*n* = 20 per group) [bFGF (−) vs. bFGF (+), 1.3 ± 0.2 vs. 2.5 ± 0.6 mm^2^; *p* = 0.00000368] (Figure 5a). A semi-quantitative evaluation of COL2 staining revealed that the bFGF (+) group had a significantly larger COL2-positive area than the bFGF (−) group (*n* = 20 per group) [bFGF (−) vs. bFGF (+), 32.9 ± 24.4% vs. 85.5 ± 14.9%; *p* = 0.0000237] (Figure 5b). In addition, the bFGF (+) group had a significantly higher sulfated glycosaminoglycan (sGAG) content than the bFGF (−) group (*n* = 8 per group) [bFGF (−) vs. bFGF (+), 10.5 ± 2.7 vs. 23.3 ± 4.5 μg/dsDNA (μg); *p* = 0.0000823] (Figure 5c). The effects of bFGF on the expression of cartilage differentiation-related genes [collagen type II alpha 1 (COL2A1), collagen type X alpha 1 (COL10A1), sex-determining region Y-box 9 (SOX9), and aggrecan (ACAN)] were analyzed by quantitative real-time polymerase chain reaction (qRT-PCR) analysis. COL2A1 gene expression in pellets from the bFGF (+) group was significantly higher than that detected in pellets from the bFGF (−) group [bFGF (−) vs. bFGF (+), 1.0 ± 0.6 vs. 2.6 ± 1.1-fold; *p* = 0.0015]. Conversely, the expression of other related genes (COL10A1, SOX9, and ACAN) was not significantly affected by bFGF administration (Figure 5d).

### 2.5. In vivo Osteochondral Repair in the Mouse Model Using Pellets Generated From the SMSCs From the bFGF (−) and bFGF (+) Groups

To assess the effect of the pretreatment with bFGF on cartilaginous regeneration of SMSC in vivo, synovial pellets were implanted into the osteochondral defects created in the knee of SCID mice. Eight weeks after implantation, no grossly visible synovitis or inflammation (Figure 6a) was observed in mouse knees in all groups, and no apparent infiltration of inflammatory cells into the repair cartilage was detected by hematoxylin and eosin (HE) staining in the two groups (Figure 6b). In both pellet-treated groups, the implanted cell aggregates were retained, and a little bony repair and a sclerotic rim formed around the implanted pellets were observed in the defects of the subchondral region.

However, the defects of the subchondral bone region were filled with repaired bone in the defect-only group. Human-vimentin-positive cells were found in the pellet implantation site in both the bFGF (−) and bFGF (+) groups, suggesting the existence of implanted human SMSCs (Figure 6c).

Moreover, the bFGF (+) group exhibited hyaline cartilage-like repair with chondrocyte-like round-shaped cells in the lacuna and abundant ECM stained with toluidine blue (Figure 7a) and Saf-O (Figure 7b); these cells also expressed COL 2 (Figure 7c). Collagen type X (COL10) expression was not observed in any of the three groups (Figure 7d). The pellet-treated groups showed significantly higher scores in the cartilage repair than the defect-only group [defect-only vs. bFGF (−), 8.8 ± 2.6 vs. 11.2 ± 1.5; *p* = 0.044; defect-only vs. bFGF (+), 8.8 ± 2.6 vs. 14.5 ± 2.6; *p* = 0.00023]. Moreover, the bFGF (+) group exhibited significantly higher scores than the bFGF (−) group (bFGF (−) vs. bFGF (+), 11.2 ± 1.5 vs. 14.5 ± 2.6; *p* = 0.0015). On the other hand, the pellet-treated groups showed significantly lower scores in the subchondral bone repair than the defect-only group [defect-only vs. bFGF (−), 7.6 ± 1.1 vs. 2.7 ± 0.5; *p* = 0.00003; defect-only vs. bFGF (+), 7.6 ± 1.1 vs. 2.8 ± 0.4; *p* = 0.00003]. In overall assessment (cartilage + subchondral bone), the bFGF (+) group exhibited significantly higher scores than the bFGF (−) group [bFGF (−) vs. bFGF (+), 13.9 ± 1.5 vs. 17.3 ± 2.6; *p* = 0.003] (Table 1).

## 3. Discussion

Fibroblast growth factors (FGFs) are expressed widely in various tissues and display various biological properties [21]. FGFs control the migration, proliferation, differentiation, and survival of various cells and affect the expression of other factors involved in the regenerative response [22]. The human–mouse FGF family comprises 22 members [23], and the bFGF (also known as FGF2) is a member of the FGF family. Most FGFs bind to heparin-sulfate proteoglycan for signaling through the fibroblast growth factor receptor (FGFR) [24] and FGFR1–4 has been identified in both humans and mice [21]. As mentioned, FGFR3 is one of the markers of mesenchymal pre-cartilaginous stem cells, and bFGF regulates undifferentiated state and multipotent differentiation of stem cells through FGFR3-mediated signaling [14,17,18]. In the present study, the FGFR3 protein was expressed in the SMSCs of all donors assessed by Western blotting, and the addition of bFGF to the growth medium changed the morphology of SMSCs and promoted its proliferation rate without altering the expression of mesenchymal surface markers (CD44, CD73, CD90, and CD105), and the chondrocyte marker (CD151). These results imply bFGF/FGFR3 could promote chondrogenic differentiation of MSCs without changing their population. SOX9 is known as a master chondrogenic factor [25] and involved in the early phase of chondrogenesis and chondrocyte proliferation [26]. bFGF upregulates SOX9 gene expression for at least first 24 h [27] and induces chondrocyte proliferation [28]. In the present study, however, SOX9 was not significantly affected by bFGF administration. This may be due to the timing of evaluation, because SOX9 gene expression was assessed at 28 days after bFGF stimulation, which may be too late to detect SOX9 upregulation by bFGF. On the other hand, bFGF controlled the maintenance of stemness in MSCs [29,30], and suppressed the cellular senescence of MSCs by downregulating transforming growth factor β2 (TGF-β2) [31]. Cellular senescence downregulates the therapeutic potential of human MSCs by reducing their migratory, homing, and immunoregulatory abilities in vivo [32,33]. The binding interaction between chondrocytes and the ECM is critical for maintaining chondrocyte activity [34]. Altogether, these findings suggest that suppressed cellular senescence and enhanced ECM production by bFGF may lead to enhanced cartilage regeneration in vivo.

Regarding the mechanism underlying the in vivo tissue repair provided by MSCs, studies have demonstrated that transplanted MSCs differentiate directly into repaired tissue [35,36]. However, other reports concluded that MSCs lead to enhanced reparative response via the secretion of growth factors, cell–cell interactions, and the release of extracellular vesicles, which contain peptides/proteins, mRNA, and microRNAs [37,38,39]. Likewise, it is unclear whether MSCs differentiate into chondrocytes or promote host-side cartilage repair [40]. Ozeki et al. reported that synovial MSCs from a green fluorescent protein-expressing rat injected into the rat’s knee joint were detectable three weeks after injection [41]. In contrast, 1,1′-dioctadecyl-3,3,3′,3′-tetramethylindocarbocyanine perchlorate-labeled pig synovial MSCs implanted into osteochondral defects of the knee were detectable at one week but could not be found at one month, after implantation [42]. In the present study, human synovial MSCs were clearly detectable 8 weeks after the operation, as assessed by human vimentin immunostaining. Moreover, synovial pellets of the bFGF (+) group exhibited a higher cartilage differentiation potential in vivo than those of the bFGF (−) group. It has been previously reported that chondrocytes overexpressing bFGF are effective in repairing osteochondral defects [43,44]. However, to the best of our knowledge, there are no reports demonstrating that adding bFGF in the culture medium enhanced the cartilage tissue repair ability of SMSCs, which is by far a simple method. Regarding tissue engineering, we developed a scaffold-free three-dimensional tissue-engineered construct (TEC) with SMSCs, which produces rich ECM [45], and reported its cartilage repair efficacy in nude rats [8] and humans [9]. In the present study, instead of TEC, we used simple synovial pellet to simplify the methods. However, unlike our previous reports using TEC [7,8], the cartilage repair ability using simple synovial pellet without bFGF was generally low. In addition, an osteosclerotic image at the boundary with the subchondral bone as well as bone remodeling delay was observed in both bFGF(−) and bFGF(+) groups. These results suggest that synovial pellet transplantation may have prevented the repair of the host-side subchondral bone in the process of producing ECM and differentiating into cartilage tissue, and further improvement in transplantation methods may be required.

This study had several limitations. First, because of the necessity of using human SMSCs, we had to employ an in vivo xenograft model using a severe combined immunodeficient (scid) mouse, in which immunological reactions may be excessively suppressed. Second, the establishment of a full cartilage defect model using a large animal model and a longer follow-up is required to evaluate the mechanical loading and long-term maintenance of the regenerated cartilage. However, to the best of our knowledge, the results of this study demonstrated for the first time that addition of bFGF to SMSC growth culture may be a simple and useful treatment option to promote cartilage regeneration both in in vitro and in vivo.

## 4. Materials and Methods

All procedures were performed in accordance with the Declaration of Helsinki. All experiments carried out were approved by the Ethical Review Board of the Osaka University Hospital (number 15409-2, approval date April 12, 2016) and the Animal Experiment Regulations of our institution. Written informed consent was obtained from all patients.

### 4.1. Patient Demographics

The human synovium was obtained from five patients (one male and four female patients; average age, 17.4 years; range, 13–22 years) at the time of arthroscopic surgery of the knee joint. Among them, four patients underwent anterior cruciate ligament reconstruction with arthroscopy and one patient underwent a second look with arthroscopy after internal fixation of osteochondritis dissecans. The average waiting period until surgery was 13.2 weeks (Table 2).

### 4.2. Isolation and Culture of Human SMSCs

The harvested synovium (average wet weight, 483 mg; range, 400–590 mg) was stored in the 0.9% sodium chloride irrigation (Baxter, Deerfield, IL, USA) at room temperature and SMSCs were isolated within at least 3 h (Figure 8a). The synovium was sheared into small fragments and digested with 0.25% trypsin/EDTA (Gibco, Carlsbad, CA, USA) for 30 min, followed by extraction of floating adipose tissues after centrifugation [46]. Non-floating tissues were resuspended in Dulbecco’s modified Eagle medium (DMEM) (Gibco) containing 10% fetal bovine serum (FBS) (Sigma-Aldrich, St. Louis, MO, USA) and supplemented with 400 U/mL of collagenase type A (Worthington, Lakewood, NJ, USA), followed by incubation at 37 °C. Three hours later, dissociated cells were collected, washed by centrifugation, and cultured as described below. Isolated SMSCs were separated into two groups: the bFGF (−) group and the bFGF (+) group. SMSCs of the bFGF (−) group were cultured in high-glucose DMEM containing 10% FBS and 1% antibiotic–antimitotic (Sigma-Aldrich) growth medium (GM), and SMSCs of the bFGF (+) group were cultured in GM with 5 ng/mL bFGF (Wako, Osaka, Japan) at 37 °C with humidified 5% CO_2_ (Figure 8b). Then, 5 ng/mL bFGF addition was adopted with reference to previous reports [16,19,47]. SMSCs were expanded as described previously and used for experiments (analyses of cell morphology, growth profile, and FACS) at passage 3 [46].

### 4.3. Cell-Proliferation Assay

The cell-proliferation capacity of SMSCs from patients #1 to #5 was assessed from passages 1 to 6 by evaluating the population doubling level (PDL). For this, 3 × 10^5^ cells/dish were seeded onto 150-mm dishes and cultured to reach approximately 90% confluence, followed by passaging after cell counting. The PDL was calculated using the following formula: PDL = log(N/N0)/log2, where N0 is the number of plated cells and N is the number of harvested cells at the time of passage. The cumulative PDL was the total PDL = Σ(PDL)n, where n is the passage number [48].

### 4.4. Western Blotting

Western blotting was conducted as previously described [49]. The primary antibodies were as follow: Anti-FGF Receptor 3 antibody (1:1000) (Cell Signaling Technology, Danvers, MA, USA, Cat# 4574) and anti-β-actin antibody (1:1000) (Cell Signaling Technology, Cat# 4967).

### 4.5. FACS Analysis

FACS was performed to evaluate the expression of four mesenchymal surface markers (CD44, CD73, CD90, and CD105), two mesenchymal negative markers (CD11b, CD271), and one chondrocyte surface marker (CD151) using a BD FACSVerse™ flow cytometer (BD Biosciences, Franklin Lakes, NJ, USA), according to the manufacturer’s protocol. SMSCs were dissociated and resuspended in 0.1% bovine serum albumin (Sigma-Aldrich) in phosphate-buffered saline (PBS (0.1% BSA–PBS)) and incubated for 30 min at room temperature with fluorescence-conjugated antibodies. The antibodies used for FACS were as follows: anti-CD11b (BioLegend, San Diego, CA, USA, Cat# 301405), anti-CD44 (BD Biosciences, Cat# 550989), anti-CD73 (BioLegend, Cat# 344004), anti-CD90 (BioLegend, Cat# 328108), anti-CD105 (BioLegend, Cat# 323208), anti-CD151 (BD Biosciences, Cat# 556057), and anti-CD271 (BD Biosciences, Cat# 557196). Isotype-specific controls and antibodies were used according to the manufacturer’s protocol. The cells were washed with 0.1% BSA–PBS twice and suspended in 0.5 mL of 0.1% BSA–PBS for analysis on a BD FACSVerse (BD Biosciences). Data retrieved from the sorting were analyzed using the BD FACSuite Software (BD Biosciences).

### 4.6. Chondrogenesis Assays

For three-dimensional pellet culture, 2 × 10^5^ cells harvested from passage 3 were centrifuged in 96 deep-well polypropylene plates (Evergreen Scientific, Vernon, CA, USA) and cultured in GM (Figure 8c). The next day, the medium was changed to 0.5 mL of chondrogenic basal medium (high-glucose DMEM (Gibco), 110 µg/mL sodium pyruvate (Gibco), 1% ITS + premix (Corning Incorporated, NY, USA), 50 µg/mL ascorbic acid-2-phosphate (Sigma-Aldrich), 40 µg/mL l-proline (Wako, Osaka, Japan), and 1% antibiotic–antimitotic) supplemented with 50 ng/mL BMP2 (Medtronic, Dublin, Ireland) [50], and 10 ng/mL TGF-β3 (Peprotech, Rocky Hill, NJ, USA). The medium was replaced twice per week and cells were maintained at 37 °C with humidified 5% CO_2_. The synovial pellets were cultured in chondrogenic basal medium for 4 weeks to induce chondrogenic differentiation and used in experiments (histological analysis, qRT-PCR analysis, and sGAG evaluation) (Figure 8d). The effects of bFGF on the expression of cartilage differentiation-related genes (COL2A1, COL10A1, SOX9, and ACAN) were analyzed by qRT-PCR. The total RNA of pellets was extracted using a Direct-ZolTM RNA kit (Zymo Research, Irvine, CA, USA) and reverse transcribed into complementary DNA using the ReverTra Ace^®^ qPCR RT Master Mix (TOYOBO, Osaka, Japan), according to the manufacturer’s protocol. qRT-PCR was performed as described previously [51]. The transcriptional levels of target genes were normalized to the level of glyceraldehyde 3-phosphate dehydrogenase (*GAPDH*) gene expression. The expression levels of each target gene were calculated using the 2^–ΔCt^ method [51]. The TaqMan assays were performed as follows: *COL2A1*, Hs00264051_m1; *COL10A1*, Hs00166657_m1; *ACAN*, Hs00153936_m1; *SOX9*, Hs01001343_g1; and *GAPDH*, Hs02758991_g1. sGAG in the pellets was measured using the Blyscan Glycosaminoglycan Assay Kit (Biocolor, Westbury, NY, USA) after lysis with papain. sGAG production was normalized to the double-stranded DNA (dsDNA) content, which was measured on a Qubit 3.0 Fluorometer (Thermo Fisher Scientific, Waltham, MA, USA) using a Qubit^TM^ dsDNA HS Assay Kit (Thermo Fisher Scientific).

### 4.7. Implantation of Synovial Pellets onto Osteochondral Defects

For implantation, 2 × 10^5^ cells harvested from passage 3 were centrifuged in 96 deep-well polypropylene plates and cultured in GM (Figure 8c), and surgery was performed on the following day. Ten-week-old male scid mice (C.B-17/Icr-scid/scidJcl; CLEA Japan, Fujinomiya, Japan) were anesthetized with an intraperitoneal injection of 0.3 mg/kg medetomidine (Nippon Zenyaku Kogyo Co., Ltd., Fukushima, Tokyo, Japan), 4.0 mg/kg midazolam (Astellas Pharma, Inc., Tokyo, Japan), and 5.0 mg/kg butorphanol (Meiji Seika Pharma Co., Ltd., Tokyo, Japan) [52]. In both knees, the femoral trochlear grooves were exposed via a medial parapatellar incision with lateral patellar dislocation, and an osteochondral defect of 1.0 mm diameter and depth was created in each knee using a micro drill (Kiso power tool, Osaka, Japan, Cat# 28854, 28855), and a microscope. A line was marked 1.0 mm from the tip of the micro drill. Subsequently, 12 knees were left empty as a defect-only group, and 26 knees were implanted with synovial pellets using a microscope. Pellets from the bFGF (−) and bFGF (+) groups were implanted randomly, and one pellet was implanted in each defect.

Eight weeks after surgery, the scid mice were sacrificed and their knees were harvested for analyses (Figure 8e).

### 4.8. Histology and Immunohistochemistry

All samples (pellets and distal femurs) were fixed in 4% paraformaldehyde and embedded in paraffin wax, followed by dehydration with an ethanol series and clearance with xylene. To evaluate osteochondral tissues, the dissected femoral ends were decalcified with 10% ethylenediaminetetraacetic acid (pH 7.4) before paraffin embedding. The samples were cut into 5-µm-thick sections and used for HE staining, toluidine blue staining, Saf-O staining, and immunohistochemistry, as described previously [51,53]. The following antibodies were used: anti-COL2 (Kyowa Pharma Chemical, Toyama, Japan, Cat# F-57, 1:500 dilution), anti-COL10 (Thermo Fisher Scientific, Cat# 14-9771-82, 1:500 dilution), and anti-human vimentin (Abcam, Cambridge, UK, Cat# ab16700, 1:100 dilution). The CSA of pellets and COL2 staining were measured automatically using an Aperio Image Scope (Leica Biosystems, Wetzlar, Germany). The sections stained with HE and Saf-O were used for histological evaluation using the O’Driscoll scoring system for cartilage repair, as described previously [54].

### 4.9. Statistical Analysis

All data were expressed as the mean ± standard deviation. Differences between two groups were assessed using the Mann–Whitney *U* test. Differences between three groups were assessed using the Kruskal–Wallis test. Significance was set at *p* < 0.05. Statistical analyses were performed using EZR (Saitama Medical Center, Jichi Medical University, Saitama, Japan) [55].

## Figures and Tables

**Figure 1 ijms-22-00300-f001:**
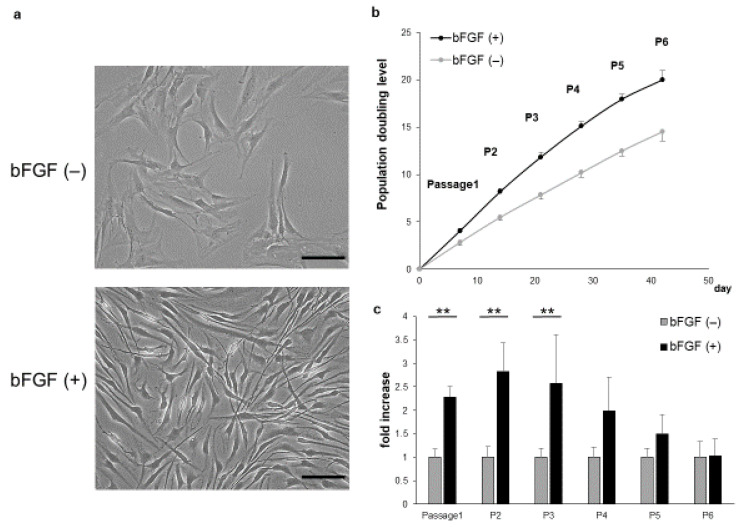
Effect of bFGF on cell proliferation and morphology. (**a**) The morphology of SMSCs was observed under a light microscope at passage 3 (bar = 100 µm). (**b**) Population doubling level of SMSCs (patients #1 to #5). (**c**) Cell counts of SMSCs in the bFGF (+) group at each passage relative to those of the bFGF (−) group (patients #1 to #5). ** *p* < 0.01. bFGF, basic fibroblast growth factor; SMSCs, synovial mesenchymal stem cells.

**Figure 2 ijms-22-00300-f002:**
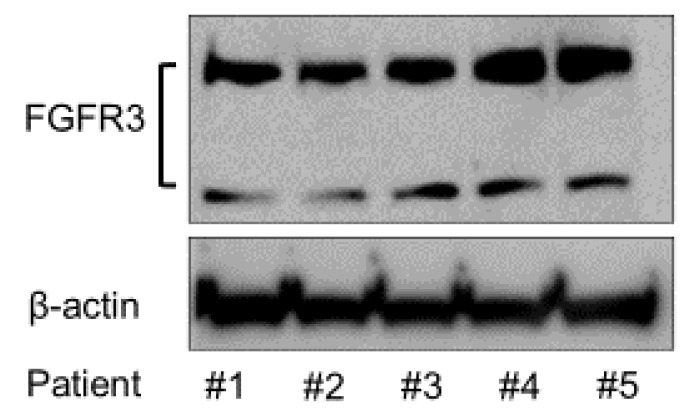
Expression of FGFR3 protein in the SMSCs of all the patients.

**Figure 3 ijms-22-00300-f003:**
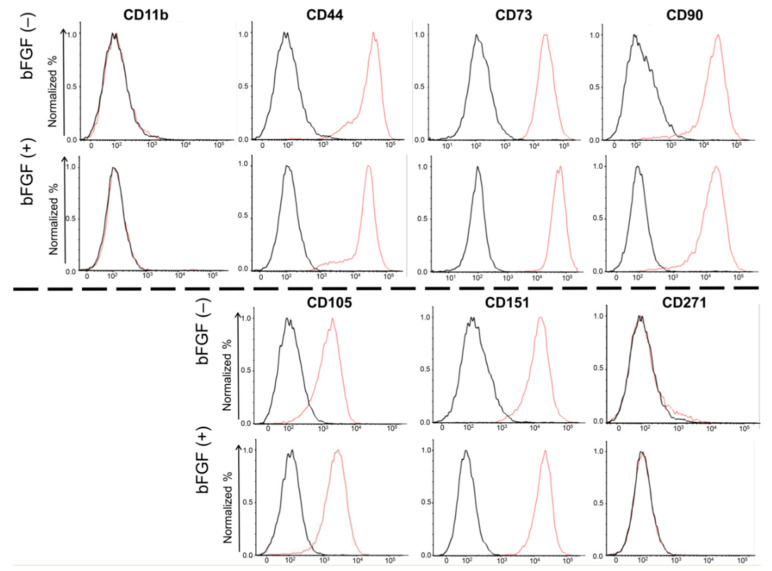
Cell surface expression of four mesenchymal surface markers (CD44, CD73, CD90, and CD105), two mesenchymal negative markers (CD11b, CD271), and one chondrocyte surface marker (CD151). SMSCs were isolated from patient #5. The results of the FACS analysis of synovial SMSCs are shown. The red line indicates the binding of the specific antibody. The black line indicates the isotypic control. SMSCs, synovial mesenchymal stem cells; FACS, fluorescence-activated cell sorting.

**Figure 4 ijms-22-00300-f004:**
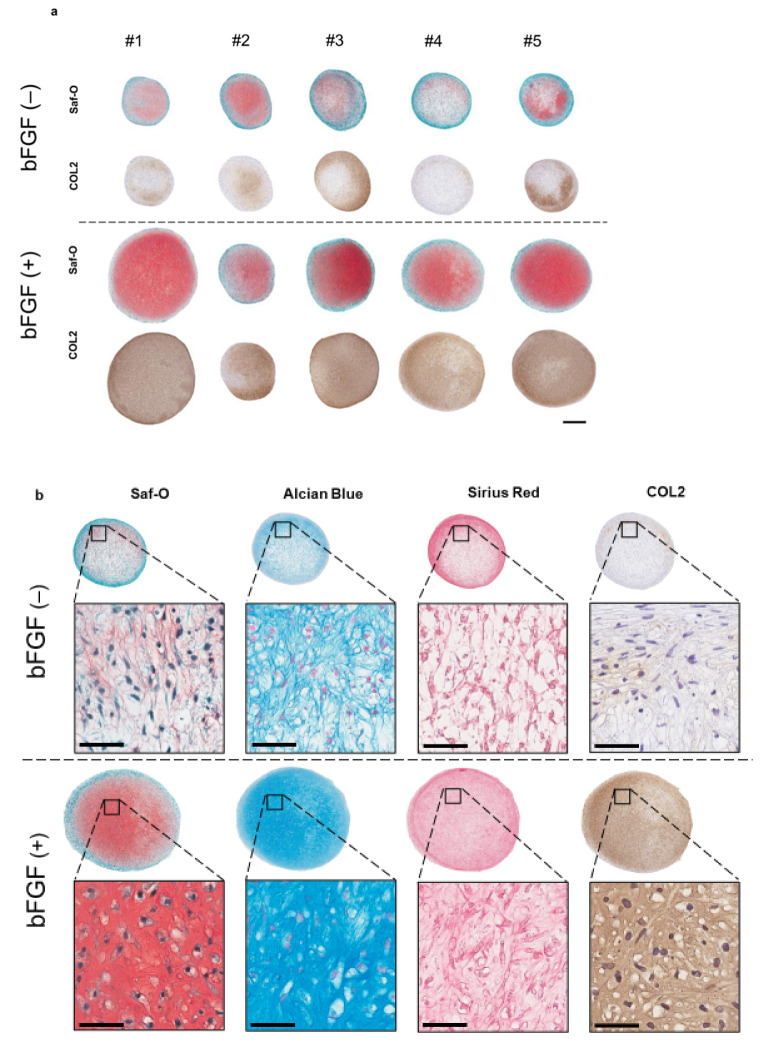
Effect of bFGF on the chondrogenesis of SMSCs in 3D chondrogenic culture: histological evaluation. Synovial pellets from both the bFGF (+) and bFGF (−) groups cultured in chondrogenic medium (without bFGF) for 4 weeks. (**a**) Images of Saf-O staining and immunostaining for COL2 (patients #1 to #5) (bar = 500 µm). (**b**) High-magnification images of Saf-O, Arcian Blue, Sirius Red, and COL2 staining (patient #4) (bar = 50 µm). bFGF, basic fibroblast growth factor; SMSCs, synovial mesenchymal stem cells; 3D, three-dimensional; Saf-O, Safranin-O; COL2, collagen type II.

**Figure 5 ijms-22-00300-f005:**
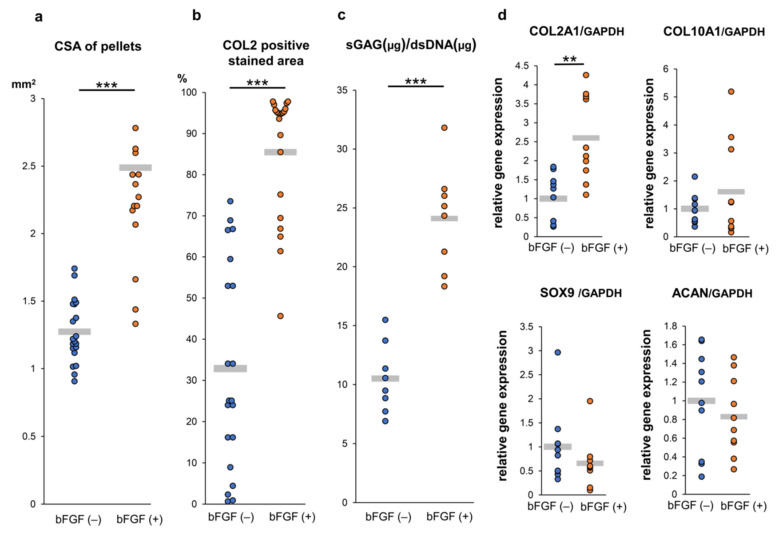
Effect of bFGF on the chondrogenesis of SMSCs in 3D chondrogenic culture: quantitative evaluation. Synovial pellets from both the bFGF (+) and bFGF (−) groups cultured in chondrogenic medium (without bFGF) for 4 weeks. (**a**) The cross-section area (CSA) of pellets (20 pellets from patients #1 to #5). (**b**) Quantitative area analysis of immunostaining for COL2 (20 pellets from patients #1 to #5). (**c**) The content of sGAG normalized to the dsDNA on day 28 (eight pellets from patients #1 and #5). (**d**) Gene expression levels in cell pellets for COL2A1, COL10A1, SOX9, and ACAN normalized to GAPDH (10 pellets from patients #3 to #5). ** *p* < 0.01; *** *p* < 0.001. bFGF, basic fibroblast growth factor; SMSCs, synovial mesenchymal stem cells; 3D, three-dimensional; CSA, cross-section area; COL2, collagen type II; sGAG, sulfated glycosaminoglycan; dsDNA, double-stranded DNA; COL2A1, collagen type II alpha 1; COL10A1, collagen type X alpha 1; SOX9, sex-determining region Y-box 9; ACAN, aggrecan; GAPDH, glyceraldehyde 3-phosphate dehydrogenase.

**Figure 6 ijms-22-00300-f006:**
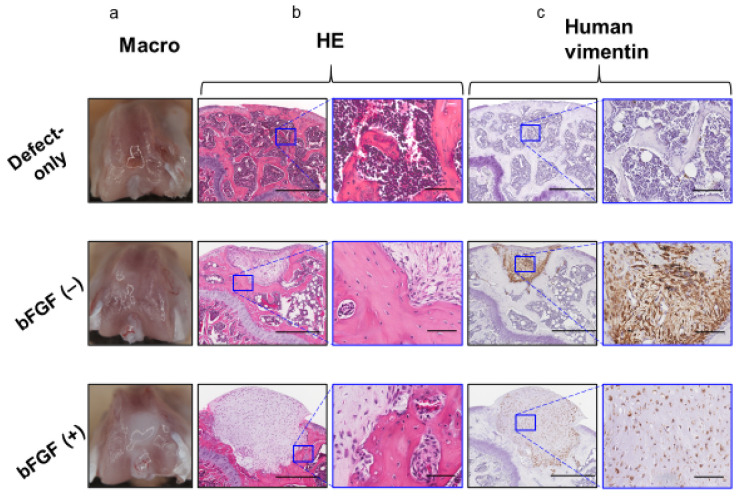
In vivo osteochondral repair in a mouse model (8 weeks after operation) (**a**) Macroscopic view of the knees at sacrifice. (**b**) Histology of the tissue stained with HE (left columns; bar = 500 µm) and the corresponding high-magnification images (right columns; bar = 100 µm). (**c**) Immunohistochemical analysis of repair tissue stained for human vimentin (left columns; bar = 500 µm) and the corresponding high-magnification images (right columns; bar = 100 µm). bFGF, basic fibroblast growth factor; HE, hematoxylin and eosin.

**Figure 7 ijms-22-00300-f007:**
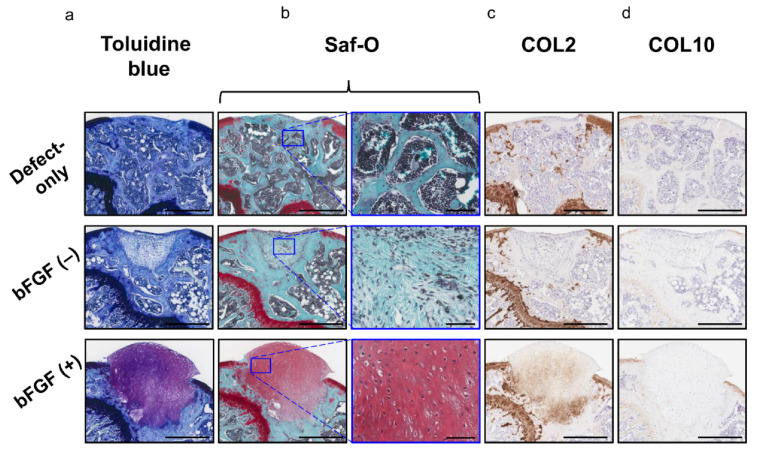
In vivo osteochondral repair in a mouse model: evaluation of chondrogenic repair (**a**) Histology of the tissue stained with toluidine blue (bar = 500 µm). (**b**) Histology of the tissue stained with Saf-O (left columns; bar = 500 µm) and the corresponding high-magnification images (right columns; bar = 100 µm). (**c**) Immunohistochemical analysis of repair cartilage stained for COL2 (bar = 500 µm). (**d**) Immunohistochemical analysis of repair tissue stained for COL10 (bar = 500 µm). bFGF, basic fibroblast growth factor; Saf-O, Safranin-O; COL2, collagen type II; COL10, collagen type X.

**Figure 8 ijms-22-00300-f008:**
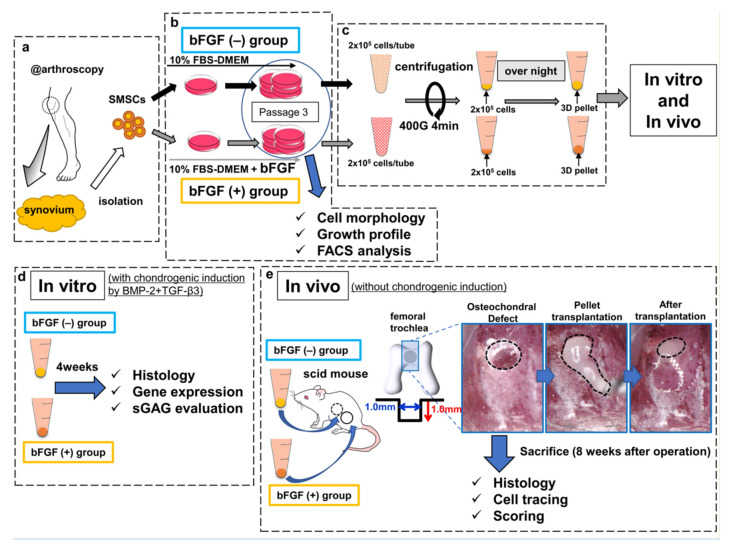
Schematic diagram of the study design. (**a**) SMSCs were isolated from the synovium obtained from human knee joints. (**b**) SMSCs were separated into the bFGF (−) and bFGF (+) groups. (**c**) For 3D pellet culture and transplantation, SMSCs harvested from passage 3 were aggregated by centrifugation. (**d**) In vitro, the pellets were cultured for 4 weeks in chondrogenic basal medium with BMP2 (50 ng/mL) and TGF-β3 (10 ng/mL), to induce chondrogenic differentiation. The histology of pellets, the expression of cartilage-differentiation-related genes, and ECM production were evaluated. (**e**) In vivo, osteochondral defects (the black dotted line in the left picture) were created in the femoral trochlea of scid mice. Synovial pellets from both groups were implanted into each knee (the black dotted line in the middle and right pictures), followed by histological evaluation (including cell tracing and scoring) 8 weeks after the operation. SMSCs, synovial mesenchymal stem cells; bFGF, basic fibroblast growth factor; 3D, three-dimensional; FBS, fetal bovine serum; DMEM, Dulbecco’s modified Eagle medium; FACS, fluorescence-activated cell sorting; BMP2, bone morphogenetic protein-2; TGF-β3, transforming growth factor-β3; ECM, extracellular matrix; sGAG, sulfated glycosaminoglycan; scid, severe combined immunodeficiency.

**Table 1 ijms-22-00300-t001:** Histological evaluation for osteochondral repair.

Histological score description	Defect only [*n* =12]	bFGF (−) [*n* =13]	bFGF (+) [*n* =13]
(possible score)			
**Cartilage repair**			
Cellular morphology (0–4)	0.7 ± 0.9	1.8 ± 0.5	2.6 ± 0.9
Safranin-O staining (0–3)	0.5 ± 0.5	0.5 ± 0.5	1.5 ± 0.7
Surface regularity (0–3)	0.8 ± 0.4	0.8 ± 0.4	1.8 ± 0.6
Structural integrity (0–2)	0.9 ± 0.3	1.1 ± 0.3	1.2 ± 0.4
Thickness (0–2)	0.4 ± 0.5	0.6 ± 0.5	1.1 ± 0.6
Bonding to adjacent cartilage (0–2)	1.4 ± 0.6	1.8 ± 0.4	1.3 ± 0.6
Hypocellularity (0–3)	0.4 ± 0.5	0.8 ± 0.4	1.0 ± 0.4
Chondrocyte clustering (0–2)	1.9 ± 0.3	1.9 ± 0.3	1.9 ± 0.3
Freedom from degeneration			
of adjacent cartilage (0–3)	1.8 ± 0.6	1.8 ± 0.4	2.1 ± 0.5
**Subtotal score (Cartilage; 0–24)**	**8.8 ± 2.6**	*** 11.2 ± 1.5**	**† 14.5 ± 2.6

**Subchondral bone repair**			
Subchondral bone alignment (0–2)	1.3 ± 0.4	0.0 ± 0.0	0.0 ± 0.0
Bone integration (0–2)	1.2 ± 0.4	0.0 ± 0.0	0.0 ± 0.0
Bone infiltration into defect area (0–2)	1.9 ± 0.3	0.7 ± 0.5	0.8 ± 0.4
Tidemark continuity (0–2)	0.0 ± 0.0	0.0 ± 0.0	0.0 ± 0.0
Cellular morphology (0–2)	1.5 ± 0.5	0.0 ± 0.0	0.0 ± 0.0
Exposure of subchondral bone (0–2)	1.8 ± 0.4	2.0 ± 0.0	2.0 ± 0.0
**Subtotal score (Subchondral bone; 0–12)**	**7.6 ± 1.1**	**** 2.7 ± 0.5**	**** 2.8 ± 0.4**


**Total Score (0–36)**	**16.3 ± 3.5**	**13.9 ± 1.5**	**† 17.3 ± 2.6**

Data were expressed as the mean ± standard deviation. * *p* < 0.05; ** *p* < 0.01 compared with the defect-only group; † *p* < 0.01 compared with the bFGF (−) group. bFGF, basic fibroblast growth factor.

**Table 2 ijms-22-00300-t002:** Patient demographics.

Patient No.	Age, y	Gender	Surgery	Waiting Period, w
#1	16	Female	ACLR	14
#2	13	Male	OCD 2nd look	15
#3	18	Female	ACLR	13
#4	18	Female	ACLR	20
#5	22	Female	ACLR	4
mean	17.4			13.2

ACLR, anterior cruciate ligament reconstruction; OCD, osteochondritis dissecans.

## Data Availability

Data available from the corresponding author on reasonable request.

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
