# Peer review of "Promoting Effect of Basic Fibroblast Growth Factor in Synovial Mesenchymal Stem Cell-Based Cartilage Regeneration"

_ijms, 2020, doi:10.3390/ijms22010300_

Round 1

Reviewer 1 Report

The manuscript entitled "Promoting effect of basic fibroblast growth factor in synovial mesenchymal stem cell-based cartilage regeneration" was interesting. As claimed by authors, the stem cells in chondrogenesis is important. The manuscript could improve by a more step-wise description of the findings and methodology used. The relevance of statements needs to be more clear for enabling readers to follow the story. Overall, major revisions are required before the manuscript will be further processed.

Line 66… FGF in chondrogenesis of SMSCs and other source-derived MSCs is well-studied in vivo and in vitro. Previously, IGF-1 and GFG-2 within cartilage defects significantly enhances the repair of full-thickness osteochondral cartilage defects (Reference…. Madry H, Orth P, Kaul G, Zurakowski D, Menger MD, Kohn D, Cucchiarini M: Acceleration of articular cartilage repair by combined gene transfer of human insulin-like growth factor I and fibroblast growth factor-2 in vivo. Arch Orthop Trauma Surg. 2010, 130: 1311-1322. 10.1007/s00402-010-1130-3.). It is well known that FGF-2 enhances early chondrogenesis. Please, show more evidence and description in the instruction to say what is new??

Line 87…. For undifferentiated Synovial fluid-derived MSCs, CD44 is specific marker. In response to FGF-2, some cells showed increased CD44 expression. I recommended to check the cell surface expression, CD44, strongly. Also, negative markers (CD11b or CD271) have to be done.

Line 108… To determine the extracellular matrix synthesis and deposition in chondrogenic differentiation, saf-o staining was performed only. However, in the figure3, the section description is very limited with low magnified images. Additional images with Sirius red staining and alcian blue staining for cartilage proteoglycan collagen will improve the author’s claim. Also, COL2 staining with FGF-2 treatment showed increased expression. It will be better if the images are in higher magnification.

Line 177….Figure 6 showed the osteochondral repair. (1) first, bFGF treatment group had large portion of saf-o staining. Interestingly, the COL2 staining portion is opposite site of saf-o stained region. Please clarify the reason.

Discussion…. The authors insisted that FGF can construct destructed tissues. However, I can not see any contractual regeneration. For cartilage degeneration, MMP13 and CD31 expression at ECM on reconstructed region are checked.

Moreover, FGF treatment increased chondrogenesis in vivo and in vitro of SMSCs. How long did you treat FGF before you started chondrogenesis?? You mentioned that we started at passage 3. It means that you cultured the SMSCs in FGF-2 treated medium for 20days (assuming from figure 1b). Right? And we didn’t add FGF in chondrogenic differentiated medium? If then, It is priming. (Reference: Delorme B, Ringe J, Pontikoglou C, Gaillard J, Langonne A, et al. (2009) Specific lineage-priming of bone marrow mesenchymal stem cells provides the molecular framework for their plasticity. Stem Cells 27: 1142–1151.) You should check the stem cell markers and pro-chondrocyte markers at passage 3, right before differentiation. FGF could trigger the pro-chondrocyte in SMSCs.

In discussion, which signalling pathway is affected by treatment of FGF to enhance the chondrogenesis?

Also, FGF2 induces chondrocyte proliferation by upregulating SOX9, as previously reported. However, in your study, SOX9 in FGF treatment group showed no significant difference. Beside PCR check, you have to check the cell surface. Please, include the explanation of SOX9 expression changes in your discussion.

Author Response

Line 66… FGF in chondrogenesis of SMSCs and other source-derived MSCs is well-studied in vivo and in vitro. Previously, IGF-1 and GFG-2 within cartilage defects significantly enhances the repair of full-thickness osteochondral cartilage defects (Reference…. Madry H, Orth P, Kaul G, Zurakowski D, Menger MD, Kohn D, Cucchiarini M: Acceleration of articular cartilage repair by combined gene transfer of human insulin-like growth factor I and fibroblast growth factor-2 in vivo. Arch Orthop Trauma Surg. 2010, 130: 1311-1322. 10.1007/s00402-010-1130-3.). It is well known that FGF-2 enhances early chondrogenesis. Please, show more evidence and description in the instruction to say what is new??

【RESPONSE】

Thank you for your comment. As reviewer pointed out, previous reports demonstrated that other methods such as 1) adding bFGF at the time of MSCs transplantation, or 2) transplanting chondrocytes overexpressing bFGF, were both effective in repairing osteochondral defects (Cucchiarini M, et al. Improved tissue repair in articular cartilage defects in vivo by rAAV-mediated overexpression of human fibroblast growth factor 2. Molecular therapy.  2005;12(2):229-238. / Filová E, et al. A cell-free nanofiber composite scaffold regenerated osteochondral defects in miniature pigs. International journal of pharmaceutics. 2013;447(1-2):139-149.). On the other hand, the present study demonstrated for the first time that adding bFGF during the proliferation culture of SMSCs (not at the implantation) significantly promoted not only cell proliferation in vitro but also the repair of osteochondral defect in vivo, which may be beneficial in both obtaining abundant cell source and also enhancing cartilage repair. According to the reviewer’s advice, we corrected our description to avoid possible misunderstandings.

Line 87…. For undifferentiated Synovial fluid-derived MSCs, CD44 is specific marker. In response to FGF-2, some cells showed increased CD44 expression. I recommended to check the cell surface expression, CD44, strongly. Also, negative markers (CD11b or CD271) have to be done.

【RESPONSE】

Thank you for your suggestion. We additionally checked the cell surface expression of CD44, CD11b, and CD271 and described in the results.

Line 108… To determine the extracellular matrix synthesis and deposition in chondrogenic differentiation, saf-o staining was performed only. However, in the figure3, the section description is very limited with low magnified images. Additional images with Sirius red staining and alcian blue staining for cartilage proteoglycan collagen will improve the author’s claim. Also, COL2 staining with FGF-2 treatment showed increased expression. It will be better if the images are in higher magnification.

【RESPONSE】

Thank you for your valuable comment. As reviewer pointed out, we modified figure 3 images and added the results of sirius red and alcian blue staining.

Line 177….Figure 6 showed the osteochondral repair. (1) first, bFGF treatment group had large portion of saf-o staining. Interestingly, the COL2 staining portion is opposite site of saf-o stained region. Please clarify the reason.

【RESPONSE】

Thank you for your comment. As reviewer pointed out, there was a dispersion in COL2 stained region and saf-o stained region. For example, in bFGF (-) of #5 patient, both regions were strongly matched. On the other hand, in- bFGF (+) of #3 patient, the margin of the pellet was strongly stained by COL2, while the center area is strongly stained by saf-o. Chondrogenesis is initialized by COL2 production and followed by aggrecan deposition (Goldring, M. B., et al. (2006). "The control of chondrogenesis." J Cell Biochem 97(1): 33-44.). In addition, the margin of the tissue is sometimes strongly stained in immunostaining. Taken together, we supposed that the reasons for this dispersion may be due to the time lag between COL2 production and aggrecan deposition, and also technical features of immunostaining. 

Discussion…. The authors insisted that FGF can construct destructed tissues. However, I can not see any contractual regeneration. For cartilage degeneration, MMP13 and CD31 expression at ECM on reconstructed region are checked.

【RESPONSE】

Thank you for important comment, and we completely agree to your suggestion. Unfortunately, we could not perform the immunohistochemistry of MMP13 and CD31 due to the limited time to revise this time. However, we would like to confirm these points in another future study. We deeply appreciate for the reviewer’s understanding.

Moreover, FGF treatment increased chondrogenesis in vivo and in vitro of SMSCs. How long did you treat FGF before you started chondrogenesis?? You mentioned that we started at passage 3. It means that you cultured the SMSCs in FGF-2 treated medium for 20days (assuming from figure 1b). Right? And we didn’t add FGF in chondrogenic differentiated medium? If then, It is priming. (Reference: Delorme B, Ringe J, Pontikoglou C, Gaillard J, Langonne A, et al. (2009) Specific lineage-priming of bone marrow mesenchymal stem cells provides the molecular framework for their plasticity. Stem Cells 27: 1142–1151.) You should check the stem cell markers and pro-chondrocyte markers at passage 3, right before differentiation. FGF could trigger the pro-chondrocyte in SMSCs.

【RESPONSE】

Thank you for your valuable comment. We cultured the SMSCs in bFGF treated medium for 20days, and we didn’t add bFGF in chondrogenic differentiated medium. As reviewer pointed out, treatment by bFGF enhanced the mRNA expression of OCT4, REX1, and NANOG of human MSC from dental pulp (Sukarawan, W., et al. (2014). "Effect of basic fibroblast growth factor on pluripotent marker expression and colony forming unit capacity of stem cells isolated from human exfoliated deciduous teeth." Odontology 102(2): 160-166.). Unfortunately, we couldn’t check the stem cell markers and pro-chondrocyte markers due to the limited time to revise this time. However, we would like to confirm these points in the future study. We deeply appreciate for the reviewer’s understanding.

In discussion, which signalling pathway is affected by treatment of FGF to enhance the chondrogenesis?

Also, FGF2 induces chondrocyte proliferation by upregulating SOX9, as previously reported. However, in your study, SOX9 in FGF treatment group showed no significant difference. Beside PCR check, you have to check the cell surface. Please, include the explanation of SOX9 expression changes in your discussion.

【RESPONSE】

Thank you for your comment. As described in the discussion, Sox9 is known as a master chondrogenic factor and involved in the early phase of chondrogenesis. bFGF upregulates SOX9 gene expression for at least first 24 hours (Murakami, S., et al. (2000). "Up-regulation of the chondrogenic Sox9 gene by fibroblast growth factors is mediated by the mitogen-activated protein kinase pathway." Proc Natl Acad Sci U S A 97(3): 1113-1118.). In the present study, however, SOX9 was not significantly affected by bFGF administration. This may be due to the timing of evaluation, because SOX9 gene expression was assessed at 28 days after bFGF stimulation, which may be too late to detect SOX9 upregulation by bFGF. On the other hand, bFGF suppresses the cellular senescence of MSCs by downregulating TGF-β2 (Ito, T., et al. (2007). "FGF-2 suppresses cellular senescence of human mesenchymal stem cells by down-regulation of TGF-beta2." Biochem Biophys Res Commun 359(1): 108-114.), and we speculated these signaling pathways may be involved. We added these descriptions in the discussion, and would like to confirm these signaling pathways in future studies.

Reviewer 2 Report

Okamura G, et al. investigated the effect of bFGF on cartilage regeneration using human synovial mesenchymal stem cell (SMSC) in vitro and in vivo. Based on their experimental findings, the authors insisted that the addition of bFGF to SMSC growth culture may be a useful treatment option to promote cartilage regeneration in vivo.

The study concept, basic experimental design and performed experiments seem to be valid and the data in the manuscript are presented clearly. I agree the basic FGF might be useful for cartilage regeneration in patients and the authors proved it in part.

However this study should contain more studies or give more information to prove the authors’ conclusion of this manuscript. Beside the study limitations described in discussion section, several concerns are also raised to this study at molecular level.

  1. In Materials and methods section, the authors obtained human synovia from five patients. How many synovia were used for this study and what was the difference among those synovia? At lease for the purpose of this study, the expression of FGFR3 in the synovial MSCs from different donors should be analyzed. The expression of FGFR3 could be analyzed by Western blotting.

  1. If possible, one or two signaling molecules downstream from bFGF/FGFR3 might be analyzed in SMSCs, especially bFGF(-) and bFGF(+) group in vitro.

  1. For addition of bFGF to SMSC growth medium, the authors add 5 ng/mL. The authors should suggest 5 ng/mL bFGF addition is optimal for chondrogenesis of SMSC. The dose-dependant reaction of bFGF might be observed and described for chondrogenic differentiation of SMSC in vitro.

Author Response

 In Materials and methods section, the authors obtained human synovia from five patients. How many synovia were used for this study and what was the difference among those synovia? At lease for the purpose of this study, the expression of FGFR3 in the synovial MSCs from different donors should be analyzed. The expression of FGFR3 could be analyzed by Western blotting.

【RESPONSE】

Thank you for your comment. We harvested several pieces of synovium (average wet weight, 483 mg; range, 400–590 mg) from each patient during endoscopic surgery. According to the reviewer’s advice, we added the detailed description about the patient demographics as a table 2 in the methods. Moreover, we checked the expression of FGFR3 by Western blotting, and described in the results.

    If possible, one or two signaling molecules downstream from bFGF/FGFR3 might be analyzed in SMSCs, especially bFGF(-) and bFGF(+) group in vitro.

【RESPONSE】

Thank you for valuable comment, and we completely agree to your suggestion. Unfortunately, we couldn’t check the signaling molecules by western blotting because of the budgetary deficit and limited time for revise. However, we would like to clarify these points in another future study. We deeply appreciate for your understanding.

    For addition of bFGF to SMSC growth medium, the authors add 5 ng/mL. The authors should suggest 5 ng/mL bFGF addition is optimal for chondrogenesis of SMSC. The dose-dependant reaction of bFGF might be observed and described for chondrogenic differentiation of SMSC in vitro.

【RESPONSE】

Thank you for important comment. It has been reported that 1-10ng/mL bFGF addition is effective for inducing chondrogenesis of MSC (Kim, J. H., et al. (2011). "Enhanced proliferation and chondrogenic differentiation of human synovium-derived stem cells expanded with basic fibroblast growth factor." Tissue Eng Part A 17(7-8): 991-1002. / Buckley, C. T. et al. (2012). "Expansion in the presence of FGF-2 enhances the functional development of cartilaginous tissues engineered using infrapatellar fat pad derived MSCs." J Mech Behav Biomed Mater 11: 102-111.). According to these previous reports and from our preliminary experiments (data not shown), we decided to add 5 ng/mL bFGF which concentration was minimum enough to induce chondrogenesis of SMSCs. We added these reference in the method.

Reviewer 3 Report

As articular cartilage is a poor self-healing tissue and there is no efficient treatment of in vivo cartilage injuries, in this study Okamura et al. investigated the effect of implantation of human synovial mesenchymal stem cells (SMSC) treated with bFGF which has been reported to promote cartilage differentiation of MSCs on in vivo cartilage regeneration. The authors examined the effect of bFGF on in vitro human SMSC proliferation and expression levels of cell surface markers, and showed that bFGF enhanced cell proliferation without affecting cell surface expression of mesenchymal markers. The authors conducted 3D pellet culture of SMSC to see the effect of bFGF on in vitro chondrogenesis, and showed that bFGF enhanced growth of 3D culture with larger COL2 expression area and sulfated glycosaminoglycan contents. The authors also found that bFGF enhanced message expression of COL2A1 in 3D culture while it did not affect expression of COL10A1, SOX9, and ACAN. Finally, the authors conducted implantation of 3D cultures treated w/o bFGF into osteochondral defects in mice, and showed that bFGF treatment greatly improved cartilage repair in terms of O’Driscoll scoring.

This is a beautiful study. Well conducted and results seems promising, although further assessment is required as the authors discussed in Discussion. I have several minor question/comments.

  1. Regarding in vivo cartilage repair experiment, expression levels of human vimentin in bFGF treated group showed relatively weaker than that in bFGF untreated group (Figure 5c). What does the meaning of expression levels of vimentin? Does it have any meaning? Does it mean maintenance of stemness or suppress senescence?
  2. In Results, it would be better to describe what experiment for what purpose the authors performed in a compressed way in each paragraph (detailed description is unnecessary, since it is well written in Materials and Methods), so that the readers can understand the flow of this study well.

Author Response

  Regarding in vivo cartilage repair experiment, expression levels of human vimentin in bFGF treated group showed relatively weaker than that in bFGF untreated group (Figure 5c). What does the meaning of expression levels of vimentin? Does it have any meaning? Does it mean maintenance of stemness or suppress senescence?

【RESPONSE】

Thank you for your comment. Human vimentin expression was evaluated to confirm the persistence of transplanted human donor cells. We suppose that the reason why vimentin expression seemed sparse in bFGF(+) compared to bFGF(-) may be due to the abundant extracellular matrix induced by bFGF, which can be considered as positive phenomenon.

  In Results, it would be better to describe what experiment for what purpose the authors performed in a compressed way in each paragraph (detailed description is unnecessary, since it is well written in Materials and Methods), so that the readers can understand the flow of this study well.

【RESPONSE】

Thank you for your comment. According to the reviewer’s advice, we described the purpose of each experiment in the result section.

Round 2

Reviewer 1 Report

The authors revised the manuscript with results. I understand that there is no enough time to go additional experiment. Modified manuscript is enough to claim the author's study.

My comments about "minor revision" are

1. discussion section with detailed descriptions about additional experimental results.

2. minor spelling check.

Author Response

  1. discussion section with detailed descriptions about additional experimental results.

【Response】

Thank you for your comment. According to the reviewer’s advice, we added the descriptions in the discussion section. Please refer to the part surrounded in yellow.

  1. minor spelling check.

【Response】

Thank you for your comment. We conducted a minor spelling check before resubmission.

Reviewer 2 Report

Although the authors did not conduct an experiment related to signal transduction of bFGF/FGFR3 in SMSCs, it could be acceptable. Because other data in the manuscript support bFGF/FGFR3 can promote chondrogenic differentiation of SMSCs.  

Author Response

Although the authors did not conduct an experiment related to signal transduction of bFGF/FGFR3 in SMSCs, it could be acceptable. Because other data in the manuscript support bFGF/FGFR3 can promote chondrogenic differentiation of SMSCs.  

【Response】

Thank you for your understanding. We would like to clarify signaling molecules downstream from bFGF/FGFR3 in another future study.